# An Effective Method for Improving Low-Frequency Response of Geophone

**DOI:** 10.3390/s23063082

**Published:** 2023-03-13

**Authors:** Kai Ma, Jie Wu, Yubo Ma, Boyi Xu, Shengyu Qi, Xiaochang Jiang

**Affiliations:** 1Department of Modern Physics, University of Science and Technology of China, Hefei 230026, China; mk2016@mail.ustc.edu.cn (K.M.);; 2State Key Laboratory of Particle Detection and Electronics, University of Science and Technology of China, Hefei 230026, China

**Keywords:** moving coil geophone, low-frequency extension, cascade correction, noise simulation

## Abstract

The natural frequency of traditional velocity sensors such as moving coil geophones limits their measurable low-frequency range, and the damping ratio affects the flatness of the sensor in the amplitude and frequency curves, resulting in variations in sensitivity over the available frequency range. In this paper, the structure and working principle of the geophone are analyzed and its dynamics are modeled. After synthesizing the negative resistance method and zero-pole compensation, two commonly adopted low-frequency extension methods, a method for improving low-frequency response, which is a series filter and a subtraction circuit to increase the damping ratio, is proposed. Applying this method to improve the low-frequency response of the JF-20DX geophone, which has a natural frequency of 10 Hz, results in a flat response to acceleration in the frequency range from 1 to 100 Hz. Both the PSpice simulation and actual measurement show a much lower noise level via the new method. Testing the vibration at 10 Hz, the new method has a 17.52 dB higher signal-to-noise ratio than the traditional zero-pole method. Both theoretical analysis and actual measurement results show that this method has a simple circuit structure, introduces less circuit noise, and has a low-frequency response improvement effect, which provides an approach for the low-frequency extension of moving coil geophones.

## 1. Introduction

Vibration signals, the reciprocating motions of an object passing through its equilibrium position, are common in nature. Vibration often has a great impact on high-precision instruments. For example, the fourth-generation synchrotron radiation light source requires monitoring of ground vibration signals in the frequency range of 1–100 Hz near the light source [1]. Seismometer measurements revealed that seismic signals generated by debris flows can have minimum frequencies as low as 2 Hz [2]. Over the past two decades, various ground vibration sensors, including hydrophones [3,4], geophones [5,6,7], seismographs [8,9,10], and fiber optic sensors [11], have been applied to detect and study ground shaking. Schimmel et al. used only one seismic sensor (geophone) collocated with an infrasound sensor, and the signal was sent to a microcontroller that ran the detection algorithm [12]. Olivier et al. adopted a 2 Hz high sensitivity geophone and low noise digital converter as a real-time seismic node [13]. Marchetti et al. [10] and Belli et al. [14] adopted seismographs (LE-3Dlite MkIII, Lennartz Electronic, Tübingen, Germany) to record ground vibration signals in the measurement range of 1–100 Hz. Chen et al. [15] applied Raspberry Shake 3D (RS3D) to detect the ground vibrations produced by an artificial debris flow.

Geophones and seismometers are the most widely applied signal receivers in vibration signal surveys, as well as accelerometers sometimes [16,17,18,19,20]. Seismometers are larger, heavier, and more expensive, but can monitor long periods and weaker signals [21]. High-frequency geophones are compact and inexpensive, but they are only suitable for detecting local and high-frequency signal components. Limited by the influence of the mechanical structure, the frequencies of such ground vibrations detected using conventional geophones are 10–100 Hz [2]. It is difficult to detect signals lower than the natural frequency, and because of the damping ratio, the sensitivity is not a stable value over the effective frequency range of the sensor. An improvement in the low-frequency response of moving coil geophones is necessary. However, it is difficult to achieve the target effect due to the limitation of its size and quality to improve the structure and performance of the sensor itself.

Commonly adopted low-frequency improvement techniques include hardware and software methods [22]. Efforts have been made to physically extend the measurement bandwidth of geophones. Pazos et al. [23] employed a capacitor in series with the shunt resistance, Oome et al. [24] applied a passive magnetic spring, and Barzilai et al. [25] operated the geophone as a closed-loop system. Beker et al. [26,27,28] employed sensor fusion strategies to construct a superior sensor. Huang et al. [11] reduced the low end of the geophone’s frequency range from 4.5 to 0.3 Hz by incorporating a current-mirror circuit. Digital or analog bandwidth extension techniques provide a means of enhancing low-frequency detection by using a dynamic sensor accuracy inverse model of the compensated geophone [29]. Analogue methods offer better real-time performance, while digital methods offer more stable compensation and immunity to interference. Digital filtering methods may be limited by the computing power of the microcontroller when considering sampling rates and multi-point measurements. Some improvements are possible with both analogue and digital methods. The analogue method is usually implemented by connecting a correction circuit in series. One of the main reasons limiting this extension method is the noise introduced by the correction circuit. After deeper extension, the noisy geophone response at low frequency causes additional interference, so the improved performance at low frequency may be canceled by the noise [30,31,32]. Usually, to reduce the noise introduced by the correction circuit, a low-noise op amp and a resistor with as small a resistance as possible are selected [33,34,35,36]. Another main reason that limits the improvement effect is the accuracy of the analog devices. Therefore, the structure of the correction circuit and the noise introduced reflect both the advantages and disadvantages of the method.

This article aims to combine two commonly adopted low-frequency improvement methods, the negative resistance method and the zero-pole compensation method [37]. The principles and ideas of the two methods are analyzed, and on this basis, a low-frequency improvement method that increases the damping ratio of the active filter in the series and the subtraction circuit is proposed. This method is relatively simple to realize the circuit; the calculation of the circuit parameters is convenient and fast, and it will not be affected by the inductance of the inner coil of the sensor. The improvement effect of this method and the response in the effective frequency range are further verified. It provides a new idea for improving the low-frequency response of the moving coil geophone.

## 2. Working Principle and Modeling of a Moving Coil Geophone

The internal structure of the moving coil detector is composed of a vibration system and a magnetic circuit system. The shell is cylindrical soft iron, the central part is a magnetic circuit system, including permanent magnets and magnetic shoes, the outer shell and magnetic shoes seal the magnetic circuit in the shell, the permanent magnets are fixed with the detector shell, and the magnetic shoes make the internal magnetic field more evenly distributed [38]. The vibrating part is located between the permanent magnet and the inner wall of the sensor shell, and an inertial system is composed of coils, brackets, and springs.

The working principle of the moving coil geophone is as follows: the sensor is fixed on the surface of the object to be measured or inserted into the ground. When the external environment of the geophone generates vibrations, the coupling problem between the sensor and the surface of the object to be measured is ignored, and it is considered to move synchronously with the object to be measured. At this time, the housing and the magnetic circuit system connected to the housing move synchronously, while the inertial system composed of springs, coils, and coil supports produces hysteresis movement due to its own inertial factors. Due to the relative motion between the coil and the permanent magnet, an induced electromotive force is generated according to Faraday’s law of electromagnetic induction. The magnitude of the induced electromotive force is proportional to the speed of relative motion. The mechanical model of the vibration system of the sensor can be simplified as a single-degree-of-freedom second-order system composed of three parts: the inertial mass M composed of the coil and the coil support, the damping C to keep the sensor stable, and the spring K to support the mass. The schematic diagram is shown in Figure 1.

From the dynamic model of the moving coil geophone, its kinematic formula can be obtained as:(1)$$m\overset{‥}{x1}+c(\dot{x1\;}-\dot{x0})+k\left(x1\;-\;x0\right)=F\left(t\right),$$
m is the mass of mass M, $\overset{‥}{x1}$ is the acceleration of the mass, c is the damping coefficient of the equivalent damper of the sensor, $\dot{x1}$ is the absolute velocity of the inertial mass, $\dot{x0}$ is the shell and the absolute velocity of the permanent magnet, k is the spring stiffness coefficient, x1 is the absolute displacement of the inertial mass, and x0 is the absolute displacement of the shell and the permanent magnet. When F(t) = 0, at this point,
(2)$$m\overset{‥}{x1}+c(\dot{x1}-\dot{\;x0})+k\left(x1\;-\;x0\right)=0,$$

Due to the relative motion of the coil inside the shell, the cutting of the magnetic induction line generates an electromotive force:(3)$$U=\mathrm{Blv}=\mathrm{Bl}(\dot{x1}-\dot{x0}),$$ B is the magnetic induction intensity of the permanent magnet, l is the equivalent length of the coil, and v is the relative speed between the mass block and the permanent magnet. The Laplace transformation of Formulas (2) and (3) gives Formulas (4) and (5), respectively.
(4)$$H_{0}(s)=\frac{X_{r}\left(s\right)}{X_{0}\left(s\right)}=\frac{-s^{2}}{s^{2}{+2\gamma }_{0}\omega_{0}s+{\omega_{0}}^{2}},$$
(5)$$U_{0}{(s)=\mathrm{BlsX}}_{r}\left(s\right){=\mathrm{GsX}}_{r}(s),$$
xr = x1 − x0, is the vibration displacement of the mass block relative to the geophone shell. X_r_(s) is the Laplace transform of xr, and G is the sensitivity of the geophone in V/(m/s). The relationship between the output of the system ($U_{0}$) and the input response ($\dot{X_{0}}$) can be obtained from Formulas (4) and (5) as Formula (6).
(6)$$U_{0}{(s)=\mathrm{GH}}_{0}{(s)\mathrm{sX}}_{0}(s)=\frac{-{\mathrm{Gs}}^{2}}{s^{2\;}{+2\gamma }_{0}\omega_{0}s+{\omega_{0}}^{2}}{\mathrm{sX}}_{0}\left(s\right)$$

The transfer function of the system can be obtained from the relationship between the input and output of the system as,
(7)$$H=\frac{-{\mathrm{Gs}}^{2}}{s^{2}{+2\gamma }_{0}\omega_{0}s+{\omega_{0}}^{2}}$$
$\omega_{0}\;$ is the natural frequency of the geophone, ${\omega_{0}}^{2}=k/m$, $\gamma_{0}$ is the geophone damping ratio, $\gamma_{0}=c/2{m\omega }_{0}$.

## 3. Series Circuit Correction Methods

### 3.1. Pole-Zero Compensation Method

The principle of the zero-pole compensation method is to cancel the pole of the original transfer function and the zero point of the compensation network through the series compensation network so that the compensated system has a new pole, thereby changing the natural frequency and damping ratio of the system. For the transfer function of the Formula (4) geophone, the target frequency after design compensation is $\omega_{1}$, and the damping ratio is $\gamma_{1}$. Then, the objective transfer function after compensation is as follows:(8)$$H=\frac{-{\mathrm{Gs}}^{2}}{s^{2}{+2\gamma }_{1}\omega_{1}s+{\omega_{1}}^{2}}$$

The zero and pole points of the transfer function change before and after compensation. Realized by a series correction network, the transfer function of the correction circuit is:(9)$$H_{0}=\frac{s^{2}{+2\gamma }_{0}\omega_{0}s+{\omega_{0}}^{2}}{s^{2}{+2\gamma }_{1}\omega_{1}s+{\omega_{1}}^{2}},$$

A change can be made to (6) to obtain:(10)$$H_{0}=\frac{s^{2}}{s^{2}{+2\gamma }_{1}\omega_{1}s+{\omega_{1}}^{2}}+\frac{{2\gamma }_{0}\omega_{0}s}{s^{2}{+2\gamma }_{1}\omega_{1}s+{\omega_{1}}^{2}}+\frac{{\omega_{1}}^{2}}{s^{2}{+2\gamma }_{1}\omega_{1}s+{\omega_{1}}^{2}}$$

Observing (7), the transfer function of the correction network can be composed of a second-order high-pass filter, a second-order bandpass filter, and a second-order lowpass filter. After the output signal of the detector passes through three filters, the outputs of the three are summed to obtain a corrected signal. The new transfer function of the system after correction using the correction network is:(11)$$H_{1}=\frac{-{\mathrm{Gs}}^{2}}{s^{2}{+2\gamma }_{0}\omega_{0}s+{\omega_{0}}^{2}}\cdot \frac{s^{2}{+2\gamma }_{0}\omega_{0}s+{\omega_{0}}^{2}}{s^{2}{+2\gamma }_{1}\omega_{1}s+{\omega_{1}}^{2}\;}=\frac{-{\mathrm{Gs}}^{2}}{s^{2}{+2\gamma }_{1}\omega_{1}s+{\omega_{1}}^{2}}$$

It can be seen from Formula (8) that after correction by the zero-pole compensation method, the new system still exhibits second-order high-pass characteristics, and the low-frequency extension can be realized by setting and selecting the natural frequency point after correction and the appropriate damping ratio.

### 3.2. Negative Resistance Method

The response function of the moving coil geophone to the vibration speed is shown in Formula (4), and its transfer function to the acceleration response is as follows:(12)$$H_{a}=\frac{-\mathrm{Gs}}{s^{2}{+2\gamma }_{0}\omega_{0}s+{\omega_{0}}^{2}},$$

The acceleration influence function conforms to the transfer function form of the second-order bandpass filter. Keeping $\omega_{0}$ unchanged and only increasing the damping ratio $\gamma_{0}$ = 0.3 to $\gamma_{1}$ = 10, then the moving coil geophone will be in the frequency range from $f_{1}{=f}_{0}\left(\gamma_{1}-\sqrt{{\gamma_{1}}^{2}-\;1}\right)=\frac{f_{0}}{{2\gamma }_{1}\;}$ to $f_{2}{=f}_{1}\left(\gamma_{1}+\sqrt{{\gamma_{1}}^{2}-\;1}\right){=2\gamma }_{1}f_{0}$, and the response to acceleration will be flat [39]. The traditional way to increase the damping ratio is to introduce negative resistance. The damping of the sensor oscillation system is mainly determined by the impedance in the coil circuit; the smaller the internal resistance of the coil circuit is, the greater the damping [40,41]. The circuit shown in Figure 2a uses an op amp to convert the positive impedance to negative, thereby introducing a negative resistance. Using the Mason rule, it is expressed as a directed graph to calculate this circuit, as shown in Figure 2b.

To do this, we introduce the following notation:(13)$$a=\frac{R_{1}}{\;R_{1}{+R}_{g}},b=\frac{R_{3}}{R_{2}{+R}_{3}},$$
then,
(14)$$\frac{V_{3}}{V_{g}}=\frac{-a/(1-a)}{1-b/(1-a)}=\frac{R_{1}\left(R_{2}{+R}_{3}\right)}{R_{1}R_{3}-R_{2}R_{g}},$$

The introduced negative resistance
(15)$$R_{\mathrm{NEG}}=-\frac{R_{1}R_{3}}{R_{2}}$$

For geophone V_g_, the load is the input impedance of R_NEG_ amplifier U_1_ at the inverting input. This impedance is subtracted from the resistance of the geophone coil R_g_, which increases the damping [39].

The disadvantage of the negative resistance method is that after the negative resistance is introduced, the impedance caused by the inductance cannot be ignored when the frequency of the system is high, resulting in an uneven acceleration response curve and distortion [37,39].

Therefore, to make the response curve flat, it is necessary to introduce negative inductance while introducing negative resistance. Song et al. [37,42,43,44] adopted genetic algorithms to repeatedly iterate and optimize the circuit topology and analog devices, including the values of resistors, capacitors, and inductors, for a negative resistance feedback network. After 47 iterations, the best curve was obtained, but at this time the topology of the correction circuit is complicated, and it is difficult to find devices with suitable values on the market for the circuit parameters, especially for the capacitance and inductance.

### 3.3. An Improved Method to Improve the Low-Frequency Response Performance of Sensors

Of the two low-frequency improvement methods mentioned above, the zero-pole compensation method modifies the transfer function of the system through a series compensation network to change the natural frequency and the damping ratio of the system. The effectiveness of the zero-pole method is influenced by the accuracy of the analogue devices, especially as the capacitors marketed have variable capacitance values and may be subject to not small deviations. The negative resistance method reduces the internal impedance of the sensor by introducing negative resistance, which in turn increases the damping ratio and flattens the original velocity response sensor in response to acceleration, hence achieving low-frequency expansion. However, due to the introduction of negative resistance, when the frequency is high, the impedance change caused by the inductance cannot be ignored, which distorts the response curve to be no longer flat.

This paper proposes an effective low-frequency response improvement method based on the two methods. The transfer function of the moving coil geophone to the acceleration response is given in Formula (12). Keeping the natural frequency of the sensor constant and increasing its damping ratio from $\gamma_{0}$ to $\gamma_{1}$, the original transfer function becomes Formula (16).
(16)$${H_{a}}^{\prime }=\frac{-\mathrm{Gs}}{s^{2}{+2\gamma }_{1}\omega_{0}s+{\omega_{0}}^{2}}$$

The transfer function of the acceleration response of the geophone can be changed by means of a series correction network. The transfer function of the correction network part is:(17)$$H_{\mathrm{compensate}}=\frac{s^{2}{+2\gamma }_{0}\omega_{0}s+{\omega_{0}}^{2}}{s^{2}{+2\gamma }_{1}\omega_{0}s+{\omega_{0}}^{2}},$$

The correction network and its transfer function change the original acceleration transfer function of the geophone.
(18)$${H_{a}}^{\prime }{=H}_{a}{\times H}_{\mathrm{compensate}}=\frac{-\mathrm{Gs}}{s^{2}{+2\gamma }_{0}\omega_{0}s+{\omega_{0}}^{2}}\cdot \frac{s^{2}{+2\gamma }_{0}\omega_{0}s+{\omega_{0}}^{2}}{s^{2}{+2\gamma }_{1}\omega_{0}s+{\omega_{0}}^{2}}=\frac{-\mathrm{Gs}}{s^{2}{+2\gamma }_{1}\omega_{0}s+{\omega_{0}}^{2}},$$

From Formula (18), there is no difference in the form of the transfer function between this correction network and the zero-pole compensation circuit, but a modification of Formula (19) can be obtained:(19)$$H_{\mathrm{compensate}}=1\;-\frac{2\left(\gamma_{1}\omega_{0\;}-{\;\gamma }_{0}\omega_{0}\right)s}{s^{2\;}{+2\gamma }_{1}\omega_{0}s+{\omega_{0}}^{2}}$$

Formula (19) changes the original transfer function. At this time, the correction network makes full use of the characteristics and transfer function of the original geophone itself. The original filter bank with an additive circuit is simplified as an all-pass filter and the subtraction of the second-order bandpass filter. The function of the correction network does not change, but the circuit structure is greatly simplified. Because the circuit structure is relatively simple, in the actual adjustment process, the improvement effect is less affected by analog devices than the zero-pole compensation method, and the calculation of circuit parameters is simpler and more convenient. Compared with the negative resistance method, which introduces negative resistance to increase the damping ratio, this method does not need to consider the influence of the inductance of the coil in the geophone on the compensated response curve. After improvement, it remains flat in the effective frequency range without distortion. For the second-order bandpass filter part in Formula (19), we choose the Sallen–Key structure, proposed by R.P. Sallen and E.L. Key of MIT Lincoln Laboratory in 1955 [45]. It has high input impedance, easy gain, and other advantages. Its basic topology is shown in Figure 3.

The transfer function of the circuit under this topology is:(20)$$H\left(s\right)=\frac{K_{f}Y_{1}Y_{2}}{\left(Y_{1}{+Y}_{2}{+Y}_{3}{+Y}_{4}\right)Y_{5}+\left[Y_{1}+\left(1\;-{\;K}_{f}\right)Y_{3}{+Y}_{4}\right]Y_{2}}$$

In the formula, $Y_{1}$~$Y_{5}$ is the complex admittance of the components at the position. For the resistance element $Y_{i}$ = 1/$R_{1}$ and the capacitance element $Y_{i}$ = s$C_{i}$, the gain of the filter circuit is $K_{f}$ = 1 + $\mathrm{R/}R_{f}$. $Y_{1}$~$Y_{5}$ can form active filter circuits of lowpass, bandpass, and high-pass by selecting appropriate resistors and capacitors, and the gain of the circuit can be changed by adjusting the resistance values of resistors R and $R_{f}$.

For a second-order bandpass filter circuit, in Formula (20), $Y_{1}$, $Y_{3}$, $Y_{5}$ take resistance, and $Y_{2}$, $Y_{4}\;$ take capacitance. The transfer function of the second-order bandpass filter circuit of unity gain is shown in Formula (21).
(21)$$H_{B}\left(s\right)=\frac{\frac{s}{R_{1}C_{2}}}{s^{2}+\left(\frac{1}{R_{1}C_{2}}+\frac{1}{R_{2}C_{2}}+\frac{1}{R_{2}C_{1}}\right)s+\frac{R_{1}{+R}_{3}}{R_{1}R_{2}R_{3}C_{1}C_{2}}}$$

The geophone is effectively heavily damped at AC frequencies because R1 is in series with the input and forms a virtual ground on the bandpass Sallen–Key. As a result, the overall noise performance of the circuit (converted to the geophone pickup side) will be lower than that of a normally used buffer amplifier at the most useful frequencies. Combine Formulas (19) and (21), hence:(22)$$\frac{1}{R_{1}C_{2}}+\frac{1}{R_{2}C_{2}}+\frac{1}{R_{2}C_{1}}{=2\gamma }_{1}\omega_{0,}$$
(23)$$\frac{R_{1}{+R}_{3}}{R_{1}R_{2}R_{3}C_{1}C_{2}}{\omega_{0}}^{2}$$

The size of 1/(R_1_C_2_) in the molecular term in Formula (21) determines the attenuation of the signal in the passband of the filter, so to avoid the attenuation of the picked-up signal as small as possible, 1/(R_1_C_2_) should be adjusted as large as possible. Yet, due to the restriction of Formula (22) in the denominator term, the value of 1/(R_1_C_2_) cannot exceed 2γ_1_ ω_0_. Therefore, we can make C_1_ = C_2_, R_2_ ≫ R_1_. In this way, Formula (22) can be approximately regarded as 1/(R_1_C_2_)$\;=\;2\gamma_{1}\;\omega_{0}$.
To reduce the introduction of circuit noise as much as possible, the resistance value of the resistor should not be too large. According to Formulas (22) and (23), based on the value of a capacitor or a resistor, the parameters of other components in the circuit can be easily calculated.

## 4. Design and Results of Sensor Low-Frequency Response Improvement

In order to verify the feasibility of the improvement method, the JF-20DX geophone was chosen as the object of the study. It was made by the French manufacturer Sercel (Nantes, France). Key parameters of the geophone are shown in the table below (Table 1).

The target frequency is set to 0.5 Hz, and the target damping ratio of zero-pole compensation is the optimal damping ratio of 0.707. Then, the target transfer function (normalized sensitivity) is:(24)$$H_{1}=\frac{s^{2}}{s^{2}+4.4422s+9.8696},$$

Bringing in the specific parameters of the JF-20DX geophone, the transfer function of the compensation network part can be calculated. The compensation network circuit diagram is shown below (Figure 4).

The essence of the correction circuit is to weigh and splice the three signals together, amplify the low frequency part, suppress the signal near the natural frequency, and keep the high frequency part unchanged to achieve the effect of flat sensitivity in the effective frequency range.

The new method of changing the damping ratio of the series network is proposed in this paper. Combined with the above theoretical analysis of the method, the objective transfer function does not change the natural frequency ω_0_ of the geophone, but only increases the damping ratio γ_0_ = 0.3 to γ_1_ = 10. It can be seen from Formula (19) that the transfer function of the correction network is obtained by subtracting two parts: the first part is an all-pass filter, and the second part is a second-order bandpass filter. The poles of the second-order bandpass filter are in the left half-plane of the s-axis, so the filter is stable. Designing this correction network only needs to implement a bandpass filter and an operational amplifier to implement a subtraction circuit to complete the correction. The damping ratio of the target system after correction is 10. Based on the above analysis, set 1/(R_1_C_2_) = 2γ_1_ω_0_. Within this setting, R_2_ ≫ R_1_. For the resistance of R_2_, it cannot be too small in order to control the reduction error of Formula (22); meanwhile, it cannot be selected too large, which may introduce considerable amounts of resistor thermal noise and op amp current source noise. To keep this balance, choose R_2_ = 1000R_1_ and take the capacitance value C_1_ = C_2_ = 10 μF; then, the resistances are R_1_ = 78 Ω and R_2_ = 78 kΩ. At this point, substituting the proportional relationship into formula (23) yields R_3_ = 2R_1_/3 = 52 Ω. The correction circuit of the new method is shown in Figure 5.

At this time, for 2γ_1_ω_0_s in the denominator term, to ensure that the signal does not attenuate during calculation, let 1/(R_1_C_2_) = 2γ_1_ω_0_ and R_2_ = 1000R_1_. Therefore, the denominator term deviates from the theoretical value and needs to be corrected in the subtraction circuit. After comparing the amplitude-frequency curve of the theoretical formula with the actual circuit amplitude-frequency curve, there is a gain deviation of 0.25 dB, so the subtraction circuit Formula (19) needs to be corrected as Formula (25).
(25)$$H_{\mathrm{compensate}}=\left(1.03-\frac{\frac{s}{R_{1}C_{2}}}{s^{2}+\left(\frac{1}{R_{1}C_{2}}+\frac{1}{R_{2}C_{2}}+\frac{1}{R_{2}C_{1}}\right)s+\frac{R_{1}{+R}_{3}}{R_{1}R_{2}R_{3}C_{1}C_{2}}}\right),$$

In (25), 1.03 is the correction for 0.25 dB gain deviation ($1.03=10^{\frac{0.25}{20}}$); here, the theoretical curve matches the circuit simulation curve. By utilizing the adjustable gain of the subtraction circuit, it is possible to increase the gain of a certain multiple to improve the sensitivity of the system. The parameters of the subtraction circuit are: R_4_ = 1 kΩ, R_5_ = 13 kΩ, R_6_ = 100 kΩ, and R_F_ = 10.3 kΩ, which can introduce a 10-fold gain to increase the overall sensitivity.

To verify the improvement effect, we also tested the sensitivity and response curve after correction under the test environment as shown in Figure 6. The computer set the vibration information, including vibration frequency and amplitude. The commands on the computer end were converted and connected to the signal generator through a controller. The signal generator created a vibration signal (usually a sinusoidal signal), and the vibration signal was amplified by a power amplifier and then connected to the vibrator. The vibration exciter generated a vibration with a fixed frequency and amplitude. A standard sensor and the system to be tested were placed on the vibration exciter. The standard sensor was used to calibrate the vibration velocity or acceleration to obtain the sensitivity of the tested system.

In the test experiments, discrete frequency points were selected, amplitudes were set, and a data acquisition system was introduced to obtain data from the standard transducer and the system under test. The frequency points were set to 0.1 Hz as the step within the range of 0.5–1 Hz, 1 Hz as the step within the range of 1–20 Hz, and 10 Hz as the step within the range of 20–100 Hz. The measured speed x response curve and frequency curve of the JF-20DX detector before and after the zero-pole method are shown in Figure 7a.

It can be seen from Figure 7a that the zero-pole compensation method achieved a better overall compensation effect. However, the attenuation of 0.5 Hz processing is −3 dB in theory, but the actual measured attenuation is approximately −5 dB, and the −3 dB attenuation point in the actual test is between 0.8 Hz and 0.9 Hz. The second is that the sensitivity fluctuates around the original natural frequency of 10 Hz of the sensor, but the overall change will not exceed 5%.

The test results of the new low-frequency response improvement method proposed in this paper are shown in Figure 7b and Figure 8a,b. Figure 7b compares the normalized amplitude-frequency curves after improvement by the two methods. Since the zero-pole method and the new method compensate for flat velocity and acceleration responses, respectively, they are normalized for comparison. The new method gives a flatter curve because the circuit structure of the method is relatively simple. The improvement is less affected by variations in the geophone parameters and the accuracy of the analogue instrument. Figure 8b gives a PSD plot of the new method and the reference sensor testing through a real clean signal.

Within 1~100 Hz, the sensitivity is approximately 194 mV/(m/s^2^), and the sensitivity attenuation is approximately −3 dB at 0.5 Hz and 200 Hz. Theoretical and actual tests of this method prove that, after the correction network, the response in the effective frequency range is flatter, the low-frequency extension can be effectively realized, and the improvement effect is good.

The phase difference curve with the reference sensor is shown in Figure 9 below. The graph includes the original geophone and the curve after improvement by both methods. The curves were smoothed by using an interpolation function.

## 5. Discussion of Noise Introduced by the Correction Circuit

The detector is compensated by means of a series circuit, and the noise distribution of each part is shown in Figure 10.

The expression of the output voltage signal is:(26)$${V_{\mathrm{out}}}^{2}=(({\left|H_{v}\right|}^{2}{P_{\mathrm{sign}}}^{2}+{E_{1}}^{2}{E_{2}}^{2}{E_{3}}^{2}){\left|H_{e}\right|}^{2}){E_{4}}^{2},$$

In Formula (26), H_v_ is the transfer function from the sensor’s speed to the point, and H_e_ is the transfer function of the correction circuit. Resistive thermal noise and mechanical thermal noise are the two main sources of noise within the geophone. The internal resistance of the sensor ranges from hundreds of ohms to several thousand ohms. For example, the internal resistance of JF-20DX is 395 ohms, which is generally much smaller than the resistance value in the correction circuit, so it can be ignored. Mechanical thermal noise E_1_ is caused by the Brownian motion of hot air molecules:(27)$${\mathrm{PSD}}_{E1}=\sqrt{16\frac{{\pi \mathrm{kT}\gamma f}_{0}}{m}}(m/s^{2}/\sqrt{\mathrm{Hz}})=\sqrt{\frac{4\mathrm{kT}\gamma }{{m\pi f}_{0}}}(m/s/\sqrt{\mathrm{Hz}})$$

Formula (27) is the power spectral density of mechanical noise, and k is the Boltzmann constant 1.38 × $10^{-23}$ J/K. T is the Kelvin temperature, usually, 293 K. It can be estimated that the power spectral density of the detector noise is on the order of $10^{-10}$, the 100 Hz bandwidth range is omitted, and the introduced noise is converted into a vibration velocity on the order of $10^{-9}$. The noise E_4_ of the acquisition circuit will not be amplified by the correction circuit. The level of noise introduced is much lower than the signal and noise amplified by the correction circuit, so it can be ignored.

From the above analysis, it can be concluded that the noise introduced by the correction circuit is very critical [21], which determines the signal-to-noise ratio of the picked-up signal and the lower limit of the vibration signal energy at different frequencies. For the zero-pole compensation circuit, the final adder adds weights to each part. The magnification of the low-frequency path in the adding circuit is the largest, so the main noise source of the zero-pole compensation circuit comes from the second-order lowpass filter circuit and the adding circuit. The second-order lowpass filter circuit noise model and the addition circuit noise model are shown in Figure 11a,b.

In this study, the main considerations of noise are the thermal noise of the resistor, the equivalent input voltage source noise, and the equivalent input current source noise of the op amp. E_Rn_ in the Figure 11 is the thermal noise of resistor R_n_, the op amp has an equivalent input voltage source noise, and both positive and negative inputs have an equivalent input current source noise.

For the new method proposed in this paper, the introduction of gain in the subtraction circuit will amplify the signal and noise at the same time. Figure 11c,d below shows the noise model diagram of the bandpass filter circuit and the noise model of the subtraction circuit. Taking Ti’s low-noise OPA209 operational amplifier as an example, the noise analysis function of the PSpice simulation software is adopted to analyze the noise of the compensation circuit in order to obtain the output noise level of the circuit.

Since different improvement methods have different amplification factors for the original signal, the results of circuit noise simulation cannot be directly compared. Therefore, the noise at the output is divided by the transfer function of the corresponding correction circuit and the transfer function of the geophone to find the noise converted to the geophone pickup vibration input and compared to the NLNM (New Low Noise Model) and NHNM (New High Noise Model) [46].

As seen in Figure 12, the noise curves for both methods are lower than the NHNM. In addition, the noise PSD of the new method is lower than the zero-pole method above the frequency of the intersection (approximately 0.8 Hz) of the two curves. This means that the new method detects vibration signals with an energy resolution below the zero-pole. As the actual noise level of the system differs from the simulation, the PSD of a vibration signal was obtained by testing different fixed frequencies of vibration in order to represent the noise level more directly. The systems compensated by both methods were measured simultaneously for the same amplitude of vibration, with vibration frequencies of 1 Hz, 10 Hz, and 20 Hz selected. Higher frequencies were not selected as the difference between acceleration and velocity of the vibration is too large at higher frequencies. The sampling rate was 1000sps, the number of acquisition points was 20,000, and a rectangular window was chosen to calculate the PSD. For visual comparison, the results of the two methods were normalized by adjusting the dominant vibration at a fixed frequency to 0 dB.

As seen in Figure 13, the noise level is generally lower with the new method, and the higher the frequency, the more significant the signal-to-noise ratio improvement is. Calculating the SNR (signal-to-noise ratio) under actual measurements, the new method outperforms the zero-pole method by 1.51 dB at 1 Hz, 17.52 dB at 10 Hz, and 21.28 dB at 20 Hz.

## 6. Conclusions

The sensitivity of moving coil vibration geophones decays rapidly below natural frequencies and their response curves fluctuate in the effective frequency band due to the damping ratio, so suitable improvement methods are required. This paper considers two effective methods of improving low-frequency response, zero-pole, and negative resistance. On the basis of the two methods, an effective method of increasing the damping ratio by using the series filter and the subtraction circuit is proposed. This method changes the original sensor that responds to speed into a response to acceleration, and the response curve is flat in the effective frequency range to achieve the effect of improving the response of the original sensor. This method, like the zero-pole, can be implemented in both analogue and digital approaches. The circuit implemented by this method is simple, which means that the improvement effect is less affected by the precision of the circuit device than the zero-pole method. Compared with the negative resistance method, this method does not need to consider the influence of inductance on the response curve.

We choose the JF-20DX geophone and design the calibration circuit using the zero-pole method and the method proposed in this paper. The zero-pole compensation method moves the natural frequency 10 Hz to approximately 0.7 Hz. The response is flat within 1–100 Hz, and the sensitivity change does not exceed 5%. The method proposed in this paper makes the response of the geophone to acceleration flat in the range of 1–100 Hz. The frequency points at which the sensitivity is attenuated by 3 dB are 0.5 Hz and 200 Hz, and the acceleration response curve at 1–100 Hz is very flat. Compared with the curve before improvement, the improvement effect is obvious. We then subtract the output noise of the circuit to the vibrating end of the geophone, compared to NLNM and NHNM. The noise level of the new method is lower than that of the zero-pole method in the range of 1–100 Hz. By measuring vibrations at the same frequency and energy and calculating the PSD and SNR of the signal, we obtain the true noise level for both methods. The SNR of the new method is 17.52 dB better than the zero-pole method for 10 Hz vibrations, for example. Both theory and experiments demonstrate the effectiveness of the method of low-frequency improvement for geophones proposed in this paper.

## Figures and Tables

**Figure 1 sensors-23-03082-f001:**
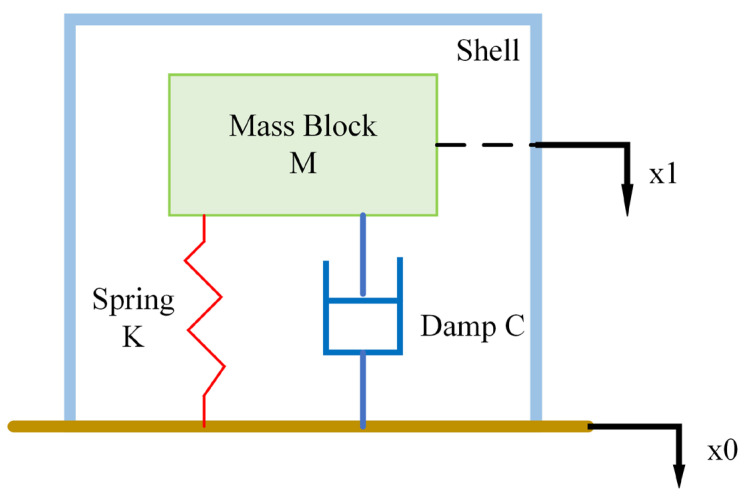
Sensor dynamics model.

**Figure 2 sensors-23-03082-f002:**
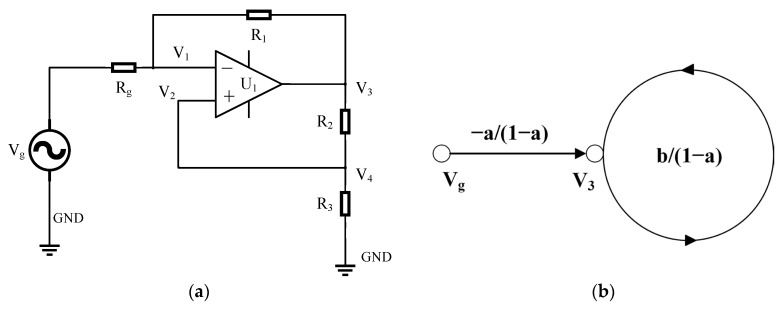
(**a**) Negative impedance inverter circuit of the sensor; (**b**) its directed graph. R_g_ is the ohmic resistance of geophone coils.

**Figure 3 sensors-23-03082-f003:**
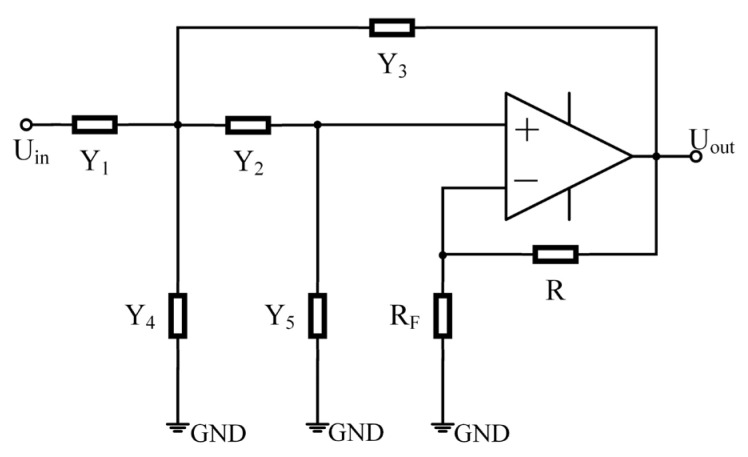
Sallen–Key second-order filter circuit topology.

**Figure 4 sensors-23-03082-f004:**
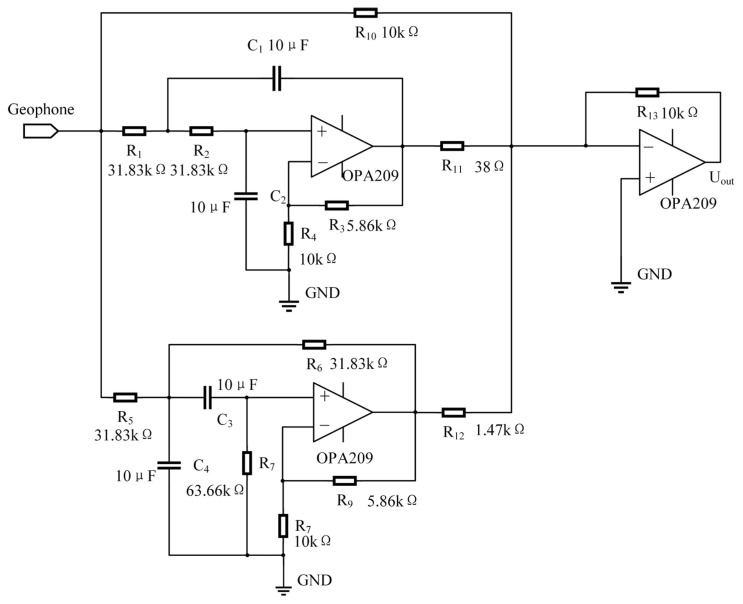
Zero-pole compensation circuit diagram.

**Figure 5 sensors-23-03082-f005:**
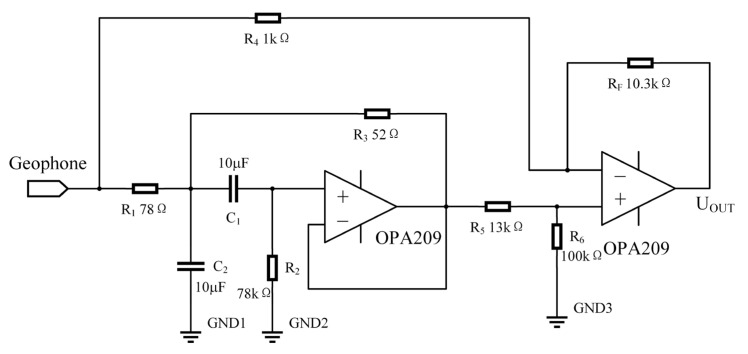
New method correction circuit diagram.

**Figure 6 sensors-23-03082-f006:**
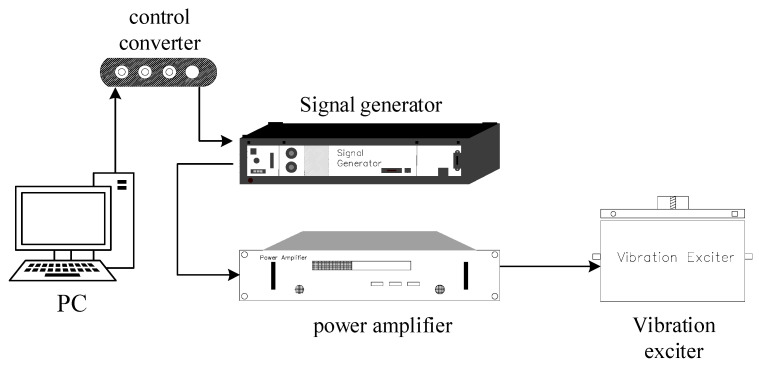
Simple schematic diagram of the test platform.

**Figure 7 sensors-23-03082-f007:**
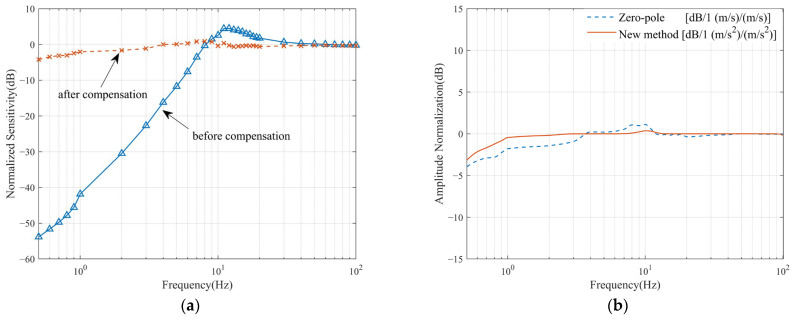
(**a**) Effect diagram of zero-pole compensation; (**b**) normalized amplitude-frequency curves of the two methods.

**Figure 8 sensors-23-03082-f008:**
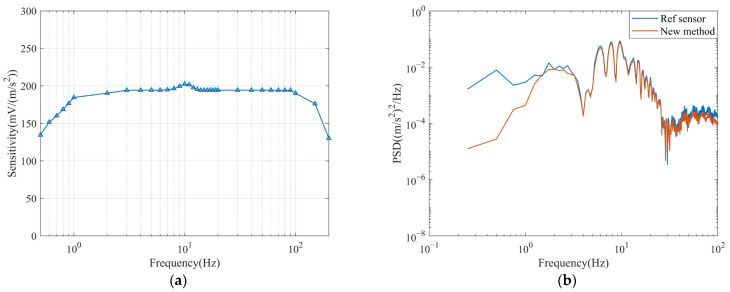
(**a**) Sensitivity curve of the system after improvement by the new method; (**b**) PSD of a segment of the real signal recorded with the reference sensor and the sensor compensated by the new method.

**Figure 9 sensors-23-03082-f009:**
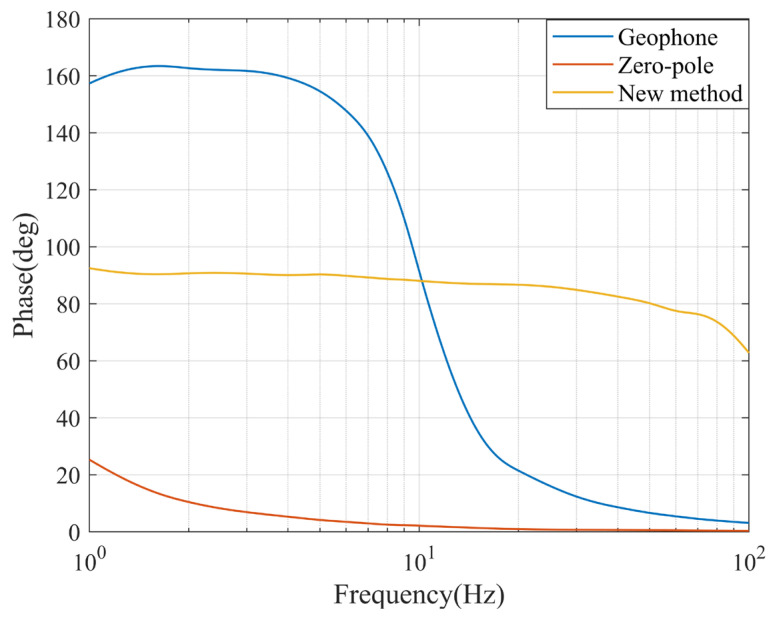
Phase difference to the reference sensor.

**Figure 10 sensors-23-03082-f010:**
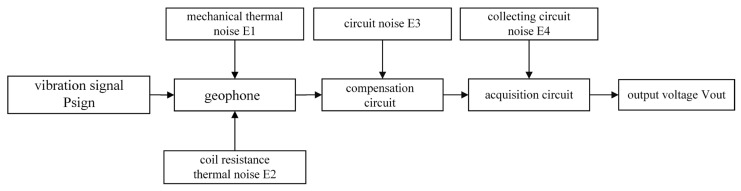
Noise Model.

**Figure 11 sensors-23-03082-f011:**
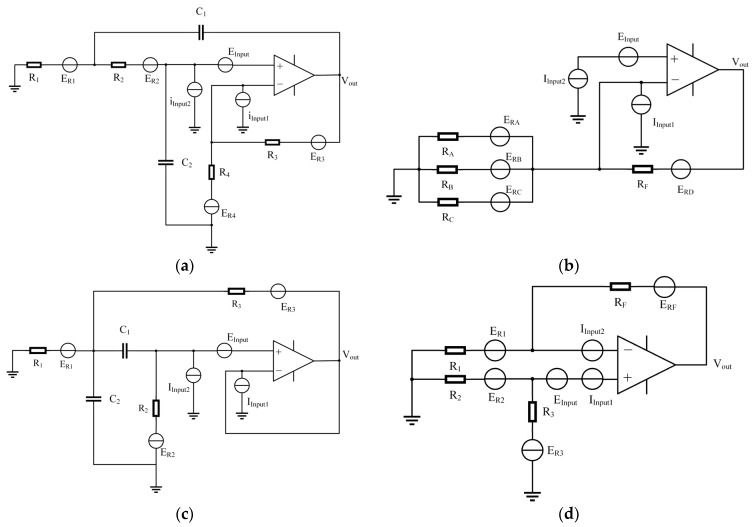
(**a**) Lowpass circuit noise equivalent model; (**b**) adding circuit noise equivalent model; (**c**) noise model of the bandpass filter circuit; (**d**) subtraction circuit noise model.

**Figure 12 sensors-23-03082-f012:**
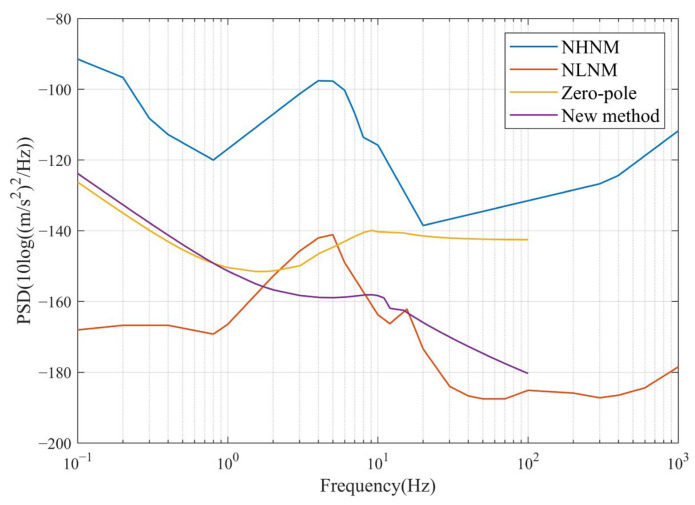
PSD of noise for both methods with NLNM and NHNM.

**Figure 13 sensors-23-03082-f013:**
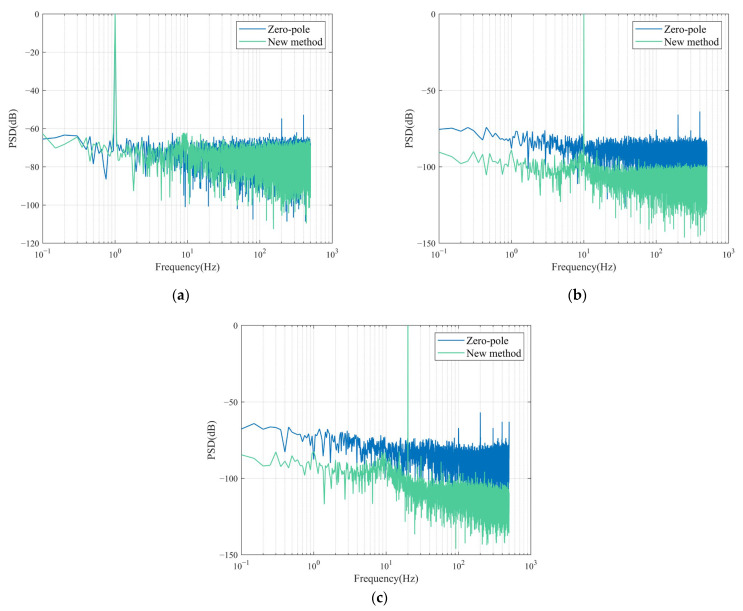
(**a**) PSD of two improvement methods for 1 Hz vibration signal; (**b**) PSD of two improvement methods for 10 Hz vibration signal; (**c**) PSD of two improvement methods for 20 Hz vibration signal.

**Table 1 sensors-23-03082-t001:** JF-20DX geophone parameter table.

JF-20DX Parameters	Value
Sensitivity	273 $\times 10^{-3}$ (V/(cm/s))
Natural frequency	10 (Hz)
Moving mass	11 (g)
Open circuit damping	0.3
Coil resistance	395 (Ω)
Inductance	50 (mH)

## Data Availability

Not applicable.

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
