# Peer review of "An Effective Method for Improving Low-Frequency Response of Geophone"

_sensors, 2023, doi:10.3390/s23063082_

Round 1
Reviewer 1 Report (New Reviewer)

Author Response
Please see the attachment.

Reviewer 2 Report (New Reviewer)
I find the term "low frequency compensation' somewhat misleading as the circuit transforms output to be proprtional to acceleration when geophones are considered to be ground velocity transducers between the low frequency resonance (10 Hz for 20DX model) and higher frequency spurious resonances.
17.52dB better is not "approximately" 17dB better
The justification for an analog citcuit versus digital post processing isn't well justified beacuse: most geophones are connected to ADC converters with over 100dB of dynamic range, using a high quality (ie linear) 10uF capacitor is almost certialy more expensive than the price of a decent 24 bit ADC. The need for two high quality amplifiers instead of one also introduces costs. High quality audio systems have struggled with the linearity and hysteritic (dielectric memory effects) performance characteristics of capacitors below 100 Hz for decades. Hence, digital compensation tends to win engeering designs.
With R1 = 78 ohm in series with input and forming a virtual ground on the band-pass Sallen key the geophone is effectively heavily damped at AC frequencies 1/(R1 C1). Thus, the overall tranducer noise performance (transfer function) of the proposed circuit would be lower at most useful frequencies than a buffer-amplifier normally used. So it is only an improvement over other analog compensation methods - this aspect should be clearer.
I feel that the principal reason to have the suggested circuit topology is in applications needing real-time feed-back and control circuits without a fast microcontroller. However, I note that most European and American engineers will still use DSP techniques for precision servo controllers.
In lines 357-358 it is claimed that Brownian motion noise is the predominant noise source within a geophone element. I understand that this is often true with MEMS sensors but the moving mass within the geophone is large enough that the intrinsic thermal vibrations should not dominante Boltzman thermal noise from damping and coil resistance (in my experience). Also in my exprience it is very rare indeed that intrinsic termal noise of a geophone is measured directly due to background earth vibrations.
Author Response
Please see the attachment.

This manuscript is a resubmission of an earlier submission. The following is a list of the peer review reports and author responses from that submission.
Round 1
Reviewer 1 Report
The authors proposed a low-frequency compensation method by increasing the damping ratio of the active filter in series and the subtraction circuit. The test results showed that the response was almost flat in the effective frequency range. It can be accepted after solving the following comments.
1. Formula (3): "v" should be expressed in terms of x1 and x0 to make formula (4) more clear.
2. Formula (4): It should be added the input and output of the system to obtain the transfer function.
3. In this paper, it mentioned in section 5: " It is calculated that the noise level of 413 the new method within 0.1-100 Hz is approximately 17 dB lower than that of the zero-pole method." However, I can't get it from figure 17. Can you explain it in detail?
Reviewer 2 Report
This paper is addressing the problem of extending the frequency response of the standard geophone, towards low frequencies. The idea is potentially significant, for e.g., seismological applications etc. The authors present two other, previously proposed methodologies, for this problem (i.e., pole zero compensation and negative resistance method) and compare them with their own method. Moreover, they present an extensive theoretical description of the moving coil geophone operation principles as well as detailed analysis of the previously proposed methods.
I think this is a good work but will need extensive rewriting before becoming suitable for publication. A large part of the text is not easily understood thus the paper needs careful reading and rewriting. There are numerous errors that prohibit clear understanding of the paper e.g. Line 246: “geographic telephone”.. what do you mean? Geophone? (see bellow for a few more). The paper has seventeen figures, must be reduced, e.g. Fig.1 is not really needed, moreover it is published elsewhere and there is no citation.
Moreover I think that you need to show to the reader, which is the noise plot of your circuit, expressed in terms of NLNM (Peterson, J., 1993. Observations and modeling of seismic background noise. U.S. Geological Survey, Open-File report 93-322, 95 pp). This will help the reader to understand the quality of your method and which is exactly the self-noise of your sensor related to NLNM.
Your method implements a velocity output sensor using a geophone, therefore you have implemented a seismometer. It would be necessary to compare this seismometer with another unit, commercially available. The comparison has to focus on signal phase difference between the reference sensor and the new proposed sensor, and to a PSD signal plot of a quiet signal part taken from the reference and the new proposed sensor. That will help the reader to better understand its performance. And note also that theoretical noise calculation is always far from the real self noise performance of any circuit, especially if mechanical elements (geophone) are included.
Also, you have selected an opamp with low noise at 1kHz but the noise density in the desired band (0.1-10Hz) is not low. Definitely you can use a better opamp with much lower noise into this band.
More corrections
Line 10: please explain better “curves not flat enough”
Line 11: use better wording “its dynamic model is established”
Line 21, 65: better use “extension” instead of “expansion”
Line 50: use better wording “is not flat enough”
Line 99: use better wording “outside world”
Line 190: use better wording of “the accuracy of the capacitors sold on the market is difficult to achieve,”. You probably mean that the capacitance of the commercial capacitors is not fixed, being affected by its tolerance, which is not small.
Line 195: replace “speed response” with “velocity response” and in general use the term “velocity”
Lines 198-200: Using negative impedances is not so difficult so you have to remove these lines [Overdamping geophones using negative impedances, Brend Ulmann, 2005] [Extending a Lippmann style seismometer’s dynamic range by using a non-linear feedback circuit, G. Romeo and G. Spinelli]
Line 204-206: use better wording.
Line 211: you need to explain better how the compensation network and its transfer function changes the original acceleration transfer function of the geophone